# Diffusion Bonding of Al-Mg-Si Alloy and 301L Stainless Steel by Friction Stir Lap Welding Using a Zn Interlayer

**DOI:** 10.3390/ma15030696

**Published:** 2022-01-18

**Authors:** Ji-Hong Dong, Hua Liu, Shu-De Ji, De-Jun Yan, Hua-Xia Zhao

**Affiliations:** 1Beijing Institute of Petrochemical Technology, Beijing 102617, China; dongjihong2021@163.com; 2Beijing Academy of Safety Engineering and Technology, Beijing 102617, China; 3AVIC Manufacturing Technology Institute, Beijing 100024, China; zhaohuaxiapaper@163.com; 4College of Aerospace Engineering, Shenyang Aerospace University, Shenyang 110136, China; LH1930375157@163.com; 5Guangdong Key Laboratory of Enterprise Advanced Welding Technology for Ships, CSSC Huangpu Wenchong Shipbuilding Company Limited, Guangzhou 510715, China

**Keywords:** friction stir lap welding, dissimilar metals, diffusion layer, Zn interlayer, mechanical properties

## Abstract

Friction stir lap welding (FSLW) is expected to join the hybrid structure of aluminum alloy and steel. In this study, the Al-Mg-Si aluminum alloy and 301L stainless steel were diffusion bonded by FSLDW with the addition of 0.1 mm thick pure Zn interlayer, when the tool pin did not penetrate the upper aluminum sheet. The characteristics of lap interface and mechanical properties of the joint were analyzed. Under the addition of Zn interlayer, the diffusion layer structure at lap interface changed from continuous to uneven and segmented. The components of the diffusion layer were more complex, including Fe-Al intermetallic compounds (IMCs), Fe-Zn IMCs and Al-Zn eutectic. The largely changed composition and thickness of uneven and segmented diffusion layer at the lap interface played a significant role in the joint strength. The tensile shear load of Zn-added joint was 6.26 kN, increasing by 41.3% than that of Zn-not-added joint. These two joints exhibited interfacial shear fracture, while the Zn interlayer enhanced the strength of diffusion bonding by extending the propagation path of cracks.

## 1. Introduction

Steels and 6xxx series aluminum alloys play a vital role in the modern transportation manufacturing [1,2]. With the increasing demand for lightweight products in rail transit, automobile and ship manufacturing, the hybrid structure of the aluminum alloy and the steel (Al/steel) is expected to replace a single steel structure [3,4]. At present, the lap structure of dissimilar metals can be joined by transient liquid phase bonding, fusion welding, brazing and solid-state welding [5,6,7,8]. However, the transient liquid phase bonding requires long time to obtain a good joining. Due to the high heat input during fusion welding, amounts of undesirable intermetallic compounds (IMCs) are easily generated at the Al/steel interface, which greatly reduces the mechanical properties of the Al/steel joint. Although brazing has the advantages of a smaller joint deformation and less IMCs generation than fusion welding, defects such as slags and porosities are easily produced because of the filler material. In order to improve the reliability of the welded joint, solid-state welding is preferred for Al/steel dissimilar metals [9].

Friction stir welding (FSW), as a mature solid-state joining technology, has achieved the joining of Al/steel [10,11]. Moreover, FSW characterized by low heat input and severe material plastic deformation can improve the mechanical properties of the Al/steel joint [12]. Compared with brazing and fusion welding, friction stir lap welding (FSLW) can effectively reduce IMCs at the lap interface of the Al/steel joint [13,14]. Kar et al. [15] investigated the effects of rotating speeds on the interfacial microstructures and mechanical properties of a 6061 Al/mild steel joint. At 1700 rpm, the joint structure in the weld containing mechanical mixing, fine particles and an intercalated interface without defects provided better mechanical properties. Moreover, Mahto et al. [16] studied the effects of welding and rotating speeds on the mechanical properties of AA6061 Al/AISI 304 steel, and the higher joint strength was obtained with the lower *w/v* due to the thickness reduction of IMCs layer.

In the study of FSW, artificial intelligence tools are used to maximize the joint strength by optimize the welding process parameter combination, such as adaptive neuro-fuzzy inference system, random vector functional link [17,18,19]. Generally speaking, how to obtain a sound welding joint with high quality needs to be investigated prior to using the artificial intelligence tools. Adding the metal interlayer at the lap interface of the dissimilar materials joint can directly affect the atom diffusion and the growth of IMCs [20,21], thereby adjusting the interface structure. In the study of Ahmed et al. [22], joints of aluminum and Zn-coated steel exhibited the higher strength at all welding parameters, showing the beneficial effect of the Zn coating on the joint strength. Zheng et al. [21] welded 6061 Al to 316 stainless steel using the Zn interlayer. Under the addition of the Zn interlayer, the thin steel-Zn mixing layer structure was discovered at the interface, and no IMCs interlayer was discovered at the interface, thereby obtaining the high-strength joint. Appropriate application of interlayer can significantly bring benefit to the microstructural and mechanical properties of the Al/steel joint [23].

Meanwhile, the relative position between the tool pin tip and the upper steel surface of the Al/steel FSLW joint is another critical factor, which attracts the attention of scholars. When the rotating tool inserts into the steel sheet, the interface geometry induced by the mechanical stirring from the rotating tool influences the loading capacity of the Al/steel lap joint [11,15,24]. In order to avoid the hook of interface, Huang et al. [25] proposed an enlarged pin head with circumferential notches to obtain the ultra-strong interface of the 6082-T6 Al and QSTE340TM steel joint. They found that the effective interfacial width was increased, and the joint strength was improved by this special tool. Haghshenas et al. [26] used friction stir-induced diffusion bonding for joining 5754 Al to DP600 steel obtained a relatively flat lap interface when the rotating tool did not contact the steel surface, because the intense thermo-mechanical deformation only occurred in the upper aluminum alloy sheet.

In this study, the diffusion bonding was adopted to realize the joining of Al/steel dissimilar metals by friction stir lap welding, and the Zn interlayer was simultaneously added to improve interface structure. With the addition of 0.1 mm thick pure Zn interlayer, the Al-Mg-Si aluminum alloy and 301L austenitic stainless steel as base metals (BMs) were used for FSLW. The effect of 0.1 mm pure Zn interlayer on lap interface structure and mechanical properties of the Al/steel joint were investigated in detail.

## 2. Materials and Methods

In the experiments, the upper and lower BMs were rolled Al-Mg-Si alloy sheets and 301L stainless steel sheets with the size of 230 mm × 120 mm × 2 mm. After X-ray fluorescence and mechanical property tests, the chemical compositions and mechanical properties of two BMs are shown in Table 1 and Table 2, respectively. Before welding, the welded region was rubbed with the 1000# sandpaper, and then the sheets were cleaned with absolute ethanol. The FSW-3LM-4012 FSW machine was used, and the sheets were fixed on a specially designed fixture before welding (Figure 1a). As shown in Figure 1c, the weld length was 200 mm, and the width of overlapping area was 50 mm. The welding tool rotated counterclockwise and moved along the weld centerline. The welding direction was parallel to the rolling direction of BMs.

The welding tool adopted in the experiments was shown in Figure 1b, and the tapered pin made by the W-Re alloys. The shoulder diameter was 15 mm and the pin length was 1.7 mm. The plunge depth of shoulder was 0.15 mm, so the distance between the pin tip and the bottom surface of aluminum alloy sheet was 0.15 mm. Therefore, the welding tool almost no wear. Besides, the rotating speed was 1600 rpm, the welding speed was 20 mm/min, and tool tilting angle was 2°, which were selected according to the reported literatures [16,27,28].

After welding, the specimens for metallographic observation and tensile shear tests were first cut by the wire-electrode cutting machine, and the cut direction was perpendicular to the welding direction. The metallographic specimens were polished by MP-1 polishing machine after metallographic sandpapers rubbing with sizes of 180#, 400#, 800#, 1200#, and 2000#. Then Keller reagent (5 mL HNO_3_ + 1.5 mL HCl + 1 mL HF + 95 mL H_2_O) was used to etched metallographic specimens. The Olympus-GX51 optical microscope was used to observe the micrographs of joints, and the microhardness was measured by the HVS-1000 tester with a 20 N load for a dwell time of 10 s. According to the standard of ISO 4136, at the room temperature, the tensile shear test was executed by the 5982 universal electronic material-testing machine with the loading speed of 2 mm/min. In order to reduce the experimental error, three tensile shear specimens were tested for each set of process parameters, and the mean values were used as the ultimate tensile shear load to evaluate the tensile shear strength of joints [16,29]. The Olympus-GX51 optical microscope was used to observe the joint fracture path. The SU3500 machine with scanning electron microscope (SEM) was used to observe the lap interface. And the element diffusion at the lap interface was also analyzed by energy disperse spectroscopy (EDS).

## 3. Results and Discussion

### 3.1. Cross Section Feature and EDS Analysis

The cross section of the joint with 0.1 mm Zn interlayer is shown in Figure 2. The material experiences the uneven thermal mechanical cycle, and therefore the aluminum alloy of the joint is divided into different zones including stir zone (SZ), thermo-mechanically affected zone (TMAZ), and heat affected zone (HAZ). The lap interface is well formed, and no volume defects such as holes appear in the SZ. Because the tool pin does not penetrate the steel sheet during welding, the obvious microstructure change does not occur in the steel surface. The bonding of lap interface mainly depends on the metallurgical bonding between aluminum alloy, steel, and Zn. During welding, the welding tool stirs and then drives the plasticized aluminum alloy to violently flow down toward the lap interface, which has an active effect on the diffusion bonding at the lap interface due to an extra pressure given by the material flow. Therefore, there are no area defects suck as kissing bond appearing in the Al/steel lap interface, as shown in Figure 2. Moreover, the lap interface of steel side is horizontal because the tool pin does not penetrate the steel sheet during welding and the steel has the relatively high melting point. Due to the mechanical stirring and friction heat from the stirring tool, the Zn material at the lap interface below the tool shoulder is liquefied and extruded, and then flows to the outside of HAZ. Meanwhile, under the direct stirring of welding tool, the aluminum material is driven to the space occupied by the original Zn interlayer. Therefore, different from the lap surface of steel side, the lap interface of aluminum side presents a curved morphology (Figure 2).

The red squares marked in Figure 2 are selected to further investigate the interface microstructure of the Al/steel joint. The same locations of Zn-not-added joints were used as the comparison. Figure 3 shows the interface microstructure and EDS analysis of region A, which is at the center of the two joints. The addition of Zn interlayer has an active effect on the interfacial bonding. In Figure 3a, the maximum thickness of diffusion layer composed of Fe-Al IMCs (abbreviated as Fe-Al IMCs layer) is 4.6 μm. The formation of IMCs at the interface can make the Al/steel joint realize metallurgical joining. However, the excessive IMCs can lead to the stress concentration, causing the initiation and propagation of cracks, which reduces the joint bearing capacity. Compared to the study of Mahto et al. [16], the maximum thickness of Fe-Al IMCs layer in this study is reduced. Movahedi et al. [30] also found that there was no adverse effect on the joint quality when the thickness of Fe-Al IMC layer was less than 2 μm. Comparing Figure 3a,b, it is known that when Zn interlayer is added, the maximum thickness of IMCs layer is reduced from 4.6 μm to 1.4 μm. This phenomenon shows that the addition of Zn interlayer is conducive to further thinning Fe-Al IMCs layer at the lap interface in the joint center of Al/steel by FSLW diffusion bonding, which is good for the improvement of joint strength when the thickness of IMCs layer is reasonable.

Figure 3c,d are EDS analysis of region A of these two joints respectively. Under the selected welding parameters, the welding heat input is adequate. Therefore, the Zn interlayer whose melting point is 420 °C is melted into liquid phase. Given that region A experiences the forging effect from the stirring tool, the liquid-phase Zn flows toward both sides along the interface gap between the two sheets, and then a new Al/steel interface is formed at region A. Under the combined action of pressure and temperature, the interdiffusion of Al and Fe atoms occurs at the interface and then Fe-Al IMCs are formed. Especially of note, combining the element analysis by EDS line scanning in Figure 3e,f, Zn content of the interface at region A of the Zn-added joint is extremely low, which reveals that a lot of Zn liquid is out of the welding center. Certainly, due to the sufficient flow of liquid Zn by the driving of the stirring tool, there is not enough time for Zn atoms to diffuse to the sides of aluminum alloy and steel. Therefore, only Fe-Al IMCs is formed at the lap interface in region A, as shown in Figure 3d,f. During diffusion bonding of Al/steel FSLW, the plasticization degree of aluminum material is higher than that of steel, so the liquid phase of Zn can carry more Al element to flow out of region A. Therefore, compared to the Zn-not-added joint (Figure 3c), region A of the Zn-added joint (Figure 3d) has the more intense diffusion degree of Fe atoms to the side of aluminum alloy.

Region B of the Zn-added joint marked in Figure 2 is selected to analyze the interface microstructure under HAZ, as shown in Figure 4. The maximum thickness of the diffusion layer in region B reaches 17.9 µm. The delamination phenomenon of diffusion layer exists. The Al-based solid solution and the granular Al-Zn eutectic are formed in the upper part of diffusion layer near the aluminum alloy side, and the lower layer near the steel side is made up of Fe-Zn IMCs, as shown in Figure 4a. Part of liquid Zn in region B under HAZ is still reserved due to the insufficient extruded effect provided by the tool, which causes a thicker diffusion layer than that of region A. Compared to region A, the diffusion layer in region B has a different structure and composition. During the FSLW process, some small size aluminum particles may separate from the aluminum alloy under the action of strong material flow, and then fall into the liquid Zn at the lap interface. According to the binary phase diagram of Al-Zn, the eutectic temperature of Al and Zn is about 381 °C, which is lower than the melting point of Zn. Moreover, the distribution density of Al element is higher than that of Zn element, as shown in Figure 4b,c. Therefore, the upper part of diffusion layer near the aluminum alloy side is mainly composed of the Al-based solid solution and the granular Al-Zn eutectic. At the same time, there are discontinuous Fe-Zn IMCs near the lap interface of steel side. Due to the barrier of Zn interlayer, it is difficult to form atomic diffusion between Fe and Al elements at region B, which makes the lower part of diffusion layer mainly form Fe-Zn IMCs rather than Fe-Al IMCs near the steel side.

Figure 5 show the magnified SEM view of region C and the corresponding area scanning results by EDS. It is found that the diffusion layer is much thicker than that of region A and B, and the maximum thickness is 228.6 µm, as shown in Figure 5a. Under the extruding effect of the stirring tool, the massive liquid Zn accumulates from region A to region C. Therefore, most Zn elements cannot completely react with Al and Fe elements. As a result, the diffusion layer is greatly thickened. Meanwhile, the delamination phenomenon of diffusion layer in region C also exists, which is similar to that of region B. According to the area scanning results by EDS in Figure 5b–d, the upper part of diffusion layer near the aluminum side mainly consists of Zn element due to the Zn accumulation. Meanwhile, there are some aluminum particles in the Zn layer of region C under the driving of the tool. The crack also appears in the diffusion layer. In addition, the Fe-Zn IMCs layer appears on the upper surface of steel sheet. The region C is still at a relatively high temperature state, and the liquid Zn inhibits the combination between Al and Fe elements, so Fe-Zn IMCs are produced. This phenomenon also shows that the effective diffusion bonding in the interface of region C is achieved. Compared with region B, region C has the smaller thickness of Fe-Zn IMCs layer because the temperature during welding decreases with increasing the distance away from the SZ.

### 3.2. Mechanical Properties

The microhardness distributions of the two joints are presented in Figure 6, and the positions of measured points are shown in Figure 2. The microhardness of steel is nearly equal to that of BM (232 HV), which reveals that the welding heat input during FSLW has almost no effect on the steel material when the tool pin not penetrating the steel sheet. In the vertical direction (Figure 6a), the middle-measured point is located at the lap interface, and the diagonal diameter of indentation is 50 μm, so the microhardness of this point has the value higher than that of aluminum BM (72 HV) and smaller than that of steel BM. In Figure 6a, three measured points are located at the SZ and have the smaller microhardness than that of aluminum BM. Figure 6b displays the microhardness distributions along the direction parallel to the lap interface. It is known that the microhardness distributions and values of the two joints are similar, and the two distributions both basically present a “W” shape. However, the HAZ of the Zn-added joint is narrower than that of the Zn-not-added joint, indicating that the Zn interlayer absorbs the welding heat during FSLW. The hardness of BM is the highest, and SZ, TMAZ and HAZ owns the smaller microhardness than BM, because the aluminum alloy used in this study is a kind of heat-treated reinforced aluminum alloy [31]. Compared with other zones, SZ has a relatively higher microhardness value due to the finer grains induced by the dynamic recrystallization, which has been reported by many researchers [11,32] and can be explained by the Hall-Petch formula [31,33].

The tensile shear load-displacement of specimens are shown in Figure 7. It can be concluded that the mean tensile load of the Zn-not-added joint is 4.43 kN, and that of the Zn-added joint is 6.26 kN, which is significantly increased by 41.3%. Compared to the tensile shear property of other studies [15,16,21,34], the tensile shear load of the Zn-added joint in this study still belongs to the high value. The addition of 0.1 mm thick Zn interlayer improves the interface bonding, and then strengthens the load resistance of the joint. The two joints both exhibit interfacial shear fracture, and their fracture paths are shown in Figure 8.

The AS of aluminum sheet is under the tensile load, and the edge of overlapping area becomes the origin of cracks. In Figure 8a,b, cracks propagate along the lap interface. The larger microhardness difference exists between the diffusion layer and steel, which is easier to induce the greater stress concentration. Therefore, according to the microscopic fracture diagram in Figure 8c,d, the IMCs layers at the bottom of diffusion layer peel off from the steel side, causing the shear failure. For the Zn-not-added joint, the Fe-Al IMCs layer at the bottom of diffusion layer peels off from the steel side. For the Zn-added joint, it can be further analyzed by combining the interface microstructures and joint tensile properties. In the center of the welding spot (region A), the Fe-Al IMCs layer still peels off from the steel side. However, in the zones which are away from the center of the joint (regions B and C) of the Zn-added joint, the compositions at the bottom of diffusion layer which is peeled off from the steel changes from Fe-Al IMCs to Fe-Zn IMCs. Compared with Fe-Al IMCs layer with steel, the Fe-Zn IMCs layer has a stronger bonding with the steel. In fact, the tensile shear load of the Zn-added joint is bigger than that of the Zn-not-added joint, as shown in Figure 7. The phenomena reveal that the modification of microstructure and composition of the Zn-added joint is beneficial to increasing the tensile shear property. In fact, the Zn thickness, the rotating speed, the welding speed, and the plunging depth all influence the mechanical property of the Al/Steel joint, and these influencing factors are interactive and complex. Therefore, the DOE method such as Zuo et al. [35] and Chitturi et al. [36] can be used to further enhance and then maximize the Al/steel joint strength in the future.

## 4. Conclusions

The Al-Mg-Si alloy and 301L steel were successfully diffusion bonded by FSLW assisted by the Zn interlayer. The relationships between the lap interface structure and the mechanical properties of the joint were discussed. The following conclusions are extracted.
(1)The diffusion layer between the aluminum and steel sheets was changed from the continuous distribution to the uneven and segmented distribution. Compared to the single Fe-Al IMCs in the diffusion layer of the Zn-not-added joint, the Fe-Zn IMCs and Al-Zn eutectic are formed in the Zn-added joint. When a 0.1 mm thick Zn interlayer was adopted, the uneven and segmented diffusion layer owned the significantly changed thickness and compositions.(2)The tensile shear load of the Zn-added joint was 6.26 kN, presenting an obvious increase of 41.3% compared with the Zn-not-added joint. These two joints presented the shear fracture along the lap interface, and the diffusion layers both peel off from the steel side. The heightened tensile shear strength by adding Zn interlayer mainly resulted from the rationally reduced and modified diffusion layer.


## Figures and Tables

**Figure 1 materials-15-00696-f001:**
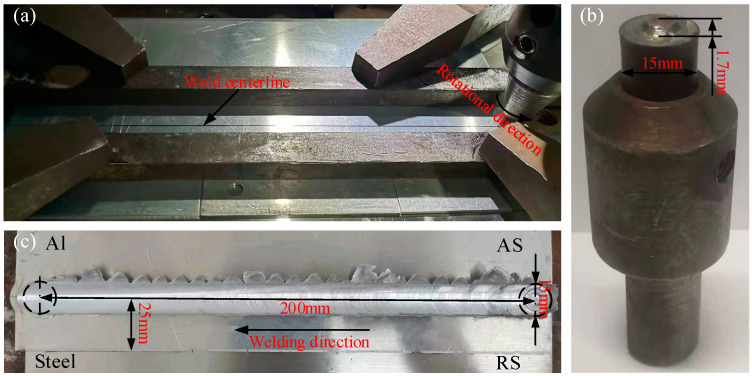
(**a**) welding process; (**b**) welding tool; (**c**) weldment.

**Figure 2 materials-15-00696-f002:**
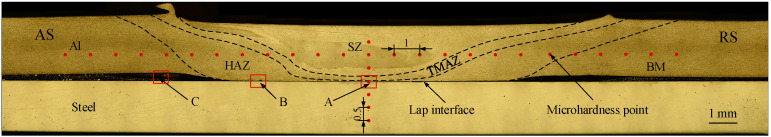
Cross section of the joint with 0.1 mm Zn interlayer.

**Figure 3 materials-15-00696-f003:**
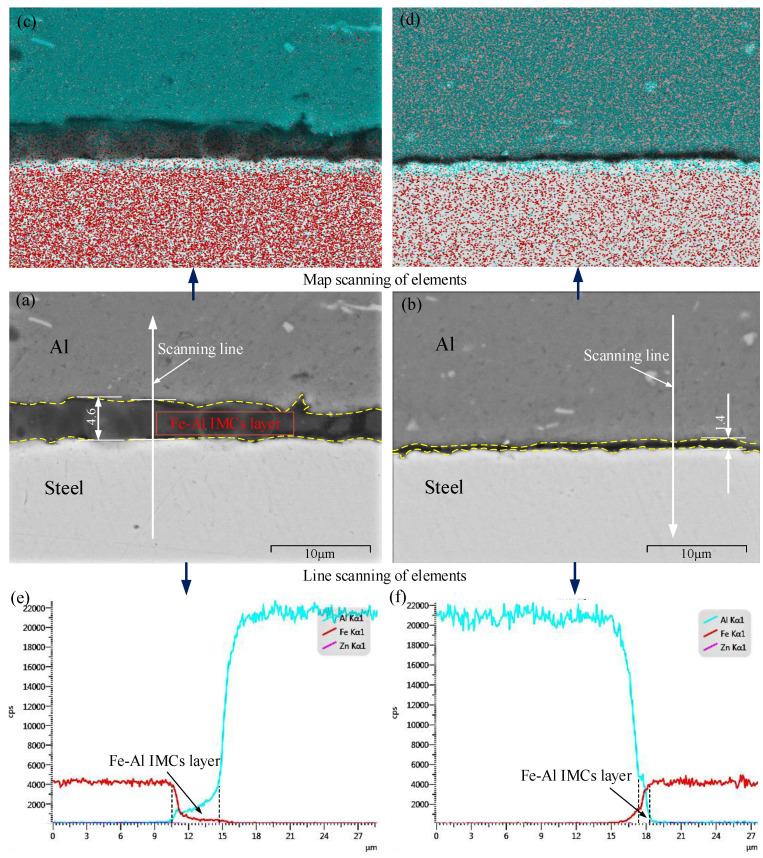
Magnified SEM views and EDS analysis results of region A: (**a**,**c**,**e**) without Zn interlayer; (**b**,**d**,**f**) with 0.1 mm Zn interlayer.

**Figure 4 materials-15-00696-f004:**
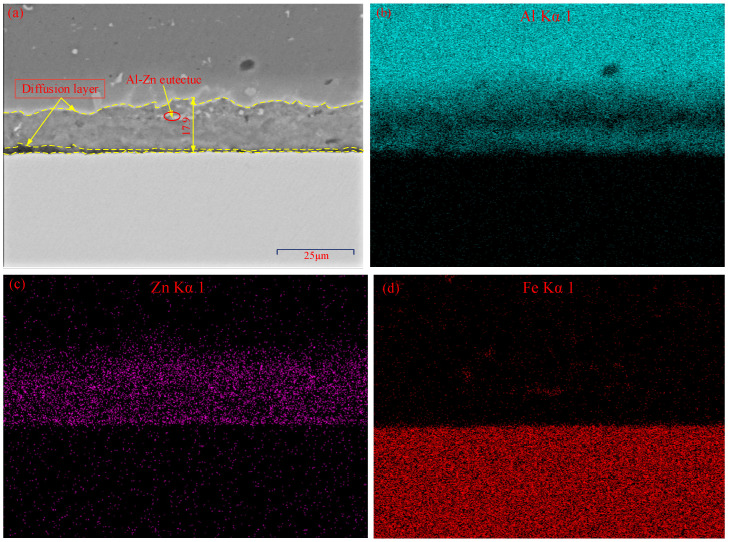
(**a**) Magnified SEM view of region B; (**b**–**d**) area scanning results of (**a**).

**Figure 5 materials-15-00696-f005:**
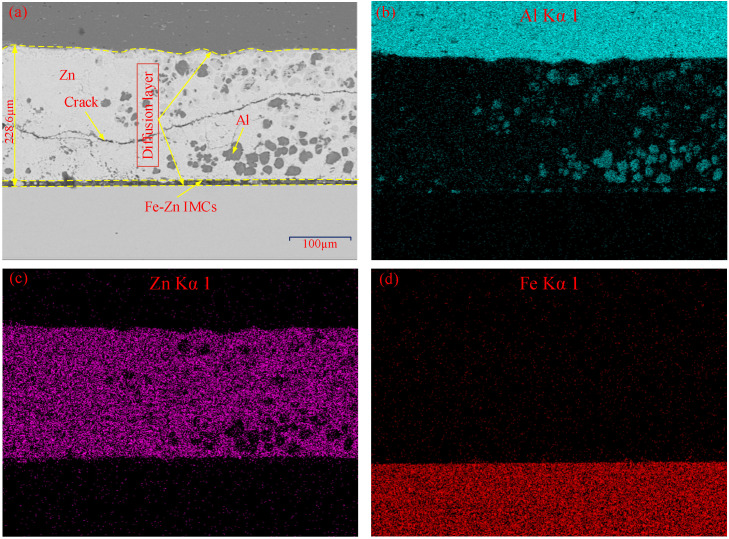
(**a**) Magnified SEM views of region C with 0.1 mm Zn interlayer; (**b**–**d**) area scanning results of (**a**).

**Figure 6 materials-15-00696-f006:**
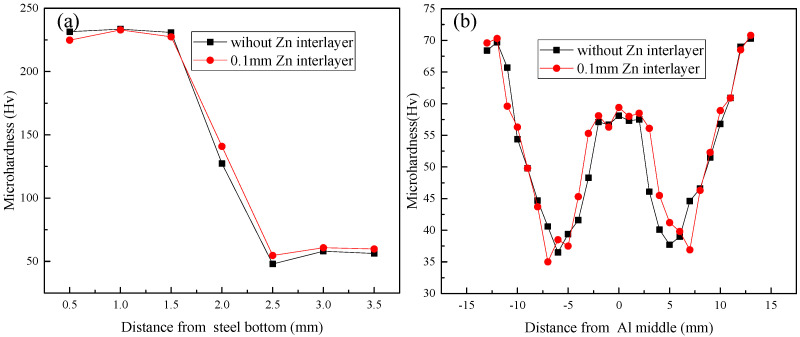
Microhardness distribution of joints along the vertical direction (**a**) and horizontal direction (**b**).

**Figure 7 materials-15-00696-f007:**
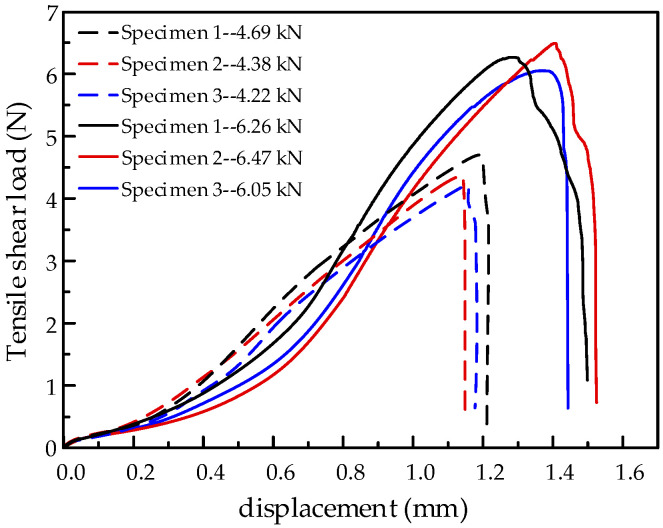
Tensile shear load-displacement of specimens.

**Figure 8 materials-15-00696-f008:**
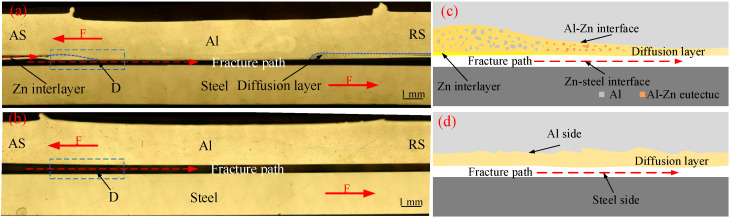
The fractured joints and the enlarged schematic views of region D marked in the fractured joint: with 0.1 mm Zn interlayer (**a**,**c**); without Zn interlayer (**b**,**d**).

**Table 1 materials-15-00696-t001:** Chemical compositions and mechanical properties of Al-Mg-Si aluminum alloy.

Chemical Composition(wt. %)								Mechanical Properties		
Si	Mg	Cr	Zn	Cu	Mn	Ti	Al	Tensile Strength	Hardness	Elongation
0.08	2.62	0.19	5.81	1.59	0.01	0.02	Bal.	355.9 MPa	99.4 Hv	7%

**Table 2 materials-15-00696-t002:** Chemical compositions and mechanical properties of 301L stainless steel.

Chemical Composition(wt. %)							Mechanical Properties		
C	N	Si	Mn	Ni	Cr	Fe	Tensile Strength	Hardness	Elongation
0.03	0.2	1.00	2.00	6.00	16.00	Bal.	864.4 Mpa	243.2 Hv	72%

## Data Availability

The raw data required to reproduce these results cannot be shared at this time as the data also forms part of an ongoing study.

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
