# Peer review of "Diffusion Bonding of Al-Mg-Si Alloy and 301L Stainless Steel by Friction Stir Lap Welding Using a Zn Interlayer"

_materials, 2022, doi:10.3390/ma15030696_

Round 1

Reviewer 1 Report

This research has attempted to use Zn layer to reinforce the weld strength of dissimilar FSW materials. The concept is not a new, many research has been done in the past in relation to introduction of Zn layer. But the material chosen for this research seems different from past literature. Hence, I suggest authors to emphasize the following.

  1. Al-Mg-Si alloy has many applications including car bodies. European industries prefer to use this alloy. Add some application of this material and need of welding Al-Mg-Si alloy with 301L steel. This will complete the research gap.
  2. I suggest that the tensile behaviour of the base metal before welding should be added to the results.
  3. The authors must report the elongation values of the samples during tensile test to help the readers to understand the phenomenon well. Add stress-strain diagram obtained from the experiments, not the drawn graph.
  4. Include the standards followed for mechanical testing.
  5. Appreciated for providing SEM image of important regions of welded samples. Please add photographic views of welded joints too in Figure 1. It will be so appreciated to have the weld process and welded samples together in a figure.
  6. Why you did not do wear test? Wear properties may also be added.
  7. Support your results with some published works. Since there are many works done in this area of research, authors may compare their results with them and highlight their findings.
  8. The following is one of the good articles on FSW. Though it is welding of similar materials, you may consider it and support your statements on process and microstructure.

    https://doi:10.1166/mex.2019.1584

  9. Check the whole manuscript for minor grammatical errors.

Author Response

Point 1: Al-Mg-Si alloy has many applications including car bodies. European industries prefer to use this alloy. Add some application of this material and need of welding Al-Mg-Si alloy with 301L steel. This will complete the research gap. 

Response 1: The contents of the application and welding need of Al-Mg-Si alloys and stainless steels has been added in Paragraph 1of Introduction. Please check!

Point 2: I suggest that the tensile behaviour of the base metal before welding should be added to the results

Response 2: The tensile properties and microhardness of Al-Mg-Si alloy BM and 301L steel BM have been measured and listed in Table 1 and Table 2 in Section 2. Please check!

Point 3: The authors must report the elongation values of the samples during tensile test to help the readers to understand the phenomenon well. Add stress-strain diagram obtained from the experiments, not the drawn graph. 

Response 3: The tensile shear load-displacement curves of tensile specimens have been added, as shown in Figure 7. According to Figure 7, the analysis of elongation values of specimens has also added in Paragraph 2 of Section 3.2. Please check!

Point 4: Include the standards followed for mechanical testing.

Response 4: The tensile tests were performed according to the standard of ISO 4136, which has been added in Paragraph 3 of Section 2. Please check!

Point 5: Appreciated for providing SEM image of important regions of welded samples. Please add photographic views of welded joints too in Figure 1. It will be so appreciated to have the weld process and welded samples together in a figure. 

Response 5: The photographic views of welded joint and the weld process have been added in Figure 1. Please check!

Point 6: Why you did not do wear test? Wear properties may also be added.

Response 6: In this study, the welding tool does not contact with the steel, so there is almost no wear. The corresponding content has been supplemented in Paragraph 2 of Section 2. Please check!

Point 7: Support your results with some published works. Since there are many works done in this area of research, authors may compare their results with them and highlight their findings.

Response 7: Compared to the results of Kar et al. [1], Mahto et al. [2], Zheng et al. [3] and Boumerzoug et al. [4], the tensile shear load of Zn-added joint in this study has been improved. The corresponding discussion has been added in Paragraph 2 of Section 3.2.

The references are as follows:

  • Kar, A.; Vicharapu, B.; Morisada, Y.; Fujii, H. Elucidation of interfacial microstructure and properties in friction stir lap welding of aluminium alloy and mild steel. Mater Charact, 2020, 168, 110572.
  • Mahto, R. P.; Kumar, R.; Pal, S. K.; Panda, S. K. A comprehensive study on force, temperature, mechanical properties, and micro-structural characterizations in friction stir lap welding of dissimilar materials (AA6061-T6 & AISI304). J Manuf Process, 2018, 31(JAN.), 624-639.
  • Zheng, Q.; Feng, X.; Shen, Y.; Huang, G.; Zhao, P. Dissimilar friction stir welding of 6061 Al to 316 stainless steel using Zn as a filler metal. J Alloys Compd, 2016, 686, 693-701.
  • Boumerzoug, Z.; Helal, Y. Friction stir welding of dissimilar materials aluminum Al6061-T6 to ultralow carbon steel. Metals, 2017, 7(2), 42.

Point 8: The following is one of the good articles on FSW. Though it is welding of similar materials, you may consider it and support your statements on process and microstructure. https://doi:10.1166/mex.2019.1584

Response 8: This above-mentioned article https://doi:10.1166/mex.2019.1584 has been considered and used to support the microhardness statement in Paragraph 1 of Section 3. Please check!

Point 9: Check the whole manuscript for minor grammatical errors.

Response 9: The grammar of the revision has been corrected, and the standard of the written English has been improved in detail. Please check!

Reviewer 2 Report

This is an interesting study, here some comments to further improve your work:

Title:
I would call this "Diffusion Bonding Al-Mg-Si Aluminum Alloy and 301L Austenitic Stainless Steel by Friction Stir Lap Welding (FSLW) using a Zn Interlayer"
simply to provide more information about the steel

or shorter "Diffusion Bonding Al-Mg-Si Aluminum Alloy and 301L Stainless Steel by Friction Stir Lap Welding (FSLW) using a Zn Interlayer"
The stainless part is really important to me since it means that the oxid layer is pretty tough and this is usually how we call 301L in the US.

Abstract:
The first sentence is overly omplicated and abstract - maybe make it simpler...

"301L steel" should always be "301L stainless steel"
What was the thickness of the Zn interlayer? was it a foil, etc. - this is relevant and can already be mentioned in the abstract to catch the readers attention
What was the purity? you used pure Zn, then it should be named as such...

line 17-18 - how were the joints analyzed, tensile testing, optical microscopy - you can already list your analyzing tools here since this tells a reader a lot about the quality of your study

Use numbers to provide quantitative results, so with 0.1 mm thickness Zn interlayer you get shear strength of 6.26 kN - how high was the load without Zn interlayer - then produce the percentage...
Mention in the abstract that you used different Zn interlayer thickness - this is very important and good - what were the thicknesses?

Manuscript:

Friction steer welding is also much faster than traditional diffusion bonding in a vacuum furnace.
https://doi.org/10.3390/met10050613 provides welding times for dissimilar bonds between stainless steels with different other materials that indicate that your process is much faster.
This should be mentioned in the introduction to show how your process is not only better than fusion welding and brazing but also diffusion bonding

What was the sandpaper roughness? did you analyze the surfaces? Cleaned with ethanol was just wiped off or did you use an ultrasonic bath?
Figure 1 indicate the shoulder size - this measurement is missing - also the total length could be interesting to better understand heat conduction
What material was used for the pin?

metallographic sandpapers until which finess?

Microhardness testing was conducted (that is great) and another analyzing method - mention it in the abstracts and methods chapter
line 138 - typo "Figlure.2" - see also Figure.5 - best check the whole manuscript quickly

Fig. & - I thought you used different Zn thickness for your analysis? so this is only with and without Zn?

The results and discssion is very good - this is how we evaluate joints as well.
So please address the other issues raised and I am confident this can be published shortly.

Author Response

Point 1: Title: I would call this "Diffusion Bonding Al-Mg-Si Aluminum Alloy and 301L Austenitic Stainless Steel by Friction Stir Lap Welding (FSLW) using a Zn Interlayer" simply to provide more information about the steel or shorter "Diffusion Bonding Al-Mg-Si Aluminum Alloy and 301L Stainless Steel by Friction Stir Lap Welding (FSLW) using a Zn Interlayer" The stainless part is really important to me since it means that the oxid layer is pretty tough and this is usually how we call 301L in the US. 

Response 1: Thank you very much for your suggestion. The title of article has been changed as "Diffusion Bonding of Al-Mg-Si Aluminium Alloy and 301L Stainless Steel by Friction Stir Lap Welding Using a Zn Interlayer" Please check!

Point 2: Abstract: The first sentence is overly complicated and abstract - maybe make it simpler...

Response 2: Thank you very much for your suggestion. The first sentence has been rewritten as "Friction stir lap welding (FSLW) is expected to join the hybrid structure of aluminum alloy and steel." Please check!

Point 3: Abstract: "301L steel" should always be "301L stainless steel" What was the thickness of the Zn interlayer? was it a foil, etc. - this is relevant and can already be mentioned in the abstract to catch the readers attention. What was the purity? you used pure Zn, then it should be named as such... 

Response 3: According to your suggestion, the type of steel, the thickness of Zn, the purity of Zn have been added in Abstract. Please check!

Point 4: Abstract: line 17-18 - how were the joints analyzed, tensile testing, optical microscopy - you can already list your analyzing tools here since this tells a reader a lot about the quality of your study.

Response 4: Thank you very much for your comment, but the analysing tools used in this study did not list in the Abstract in the revision because the words number of Abstract was limited by the journal of Materials. Certainly, the test methods and analysis methods have been described in detail in Section 2. Please check!

Point 5: Abstract: Use numbers to provide quantitative results, so with 0.1 mm thickness Zn interlayer you get shear strength of 6.26 kN - how high was the load without Zn interlayer - then produce the percentage... Mention in the abstract that you used different Zn interlayer thickness - this is very important and good - what were the thicknesses?

Response 5: The tensile shear load (4.41 kN) of Zn-not-added joint has been added in Abstract. This study aimed at the comparison and analysis of Zn-added joint and Zn-not-added joint, and the 0.1mm thick pure Zn foil was used as the interlayer. We are sorry for our inappropriate expression of different thickness of Zn interlayer in Abstract.

Point 6: Friction steer welding is also much faster than traditional diffusion bonding in a vacuum furnace. https://doi.org/10.3390/met10050613 provides welding times for dissimilar bonds between stainless steels with different other materials that indicate that your process is much faster. This should be mentioned in the introduction to show how your process is not only better than fusion welding and brazing but also diffusion bonding.

Response 6: The reference listed in the following has been supplemented into Paragraph 1 of Section 1. Please check.

[1]  Alhazaa, A.; Haneklaus, N. Diffusion bonding and transient liquid phase (TLP) bonding of type 304 and 316 austenitic stainless steela review of similar and dissimilar material joints. Metals, 2020, 10(5), 613.

Point 7: What was the sandpaper roughness? did you analyze the surfaces? Cleaned with ethanol was just wiped off or did you use an ultrasonic bath? Figure 1 indicate the shoulder size - this measurement is missing - also the total length could be interesting to better understand heat conduction What material was used for the pin?

Response 7: There were attached sundries on the surface of Al and steel sheets, so the welding surfaces of Al and steel sheets were polished with 1000# of sandpaper, and cleaned with alcohol before welding. Both sheets were naturally dried after the pure alcohol was used. Moreover, the size and material of tool pin have been added in Paragraph 2 of Section 2. Please check!

Point 8: Metallographic sandpapers until which finess?

Response 8: The sizes of metallographic sandpapers are 180#, 400#, 800#, 1200# and 2000#, respectively, which has been added in Paragraph 3 of Section 2.

Point 9: Microhardness testing was conducted (that is great) and another analyzing method - mention it in the abstracts and methods chapter line 138 - typo "Figlure.2" - see also Figure.5 - best check the whole manuscript quickl.

Response 9: We have checked the whole manuscript, and the expression form of "Figure x" has been unified and corrected in the revision. Please check!

Point 10: Fig. & - I thought you used different Zn thickness for your analysis? so this is only with and without Zn?

Response 10: We feel sorry for the inappropriate expression in the manuscript. In fact, this study aimed at the influence mechanism of the Zn interlayer on the microstructure and mechanical properties of Al/steel joint. The joint with 0.1mm pure Zn foil and the joint without Zn foil were welded by FSLW, and the microstructures at the lap interface of these two joints were compared and analyzed in detail. In addition, the microhardness, and tensile properties of these two joints were measured. The effects of Zn interlayer on the microstructure evolution and mechanical properties have been analysed and concluded.

Reviewer 3 Report

In this study, the Al- 15Mg-Si aluminum alloy and 301L steel were diffusion bonded by FSLW with the addition of Zn in- terlayer, when the tool pin did not penetrate the upper aluminum sheet. The formation of lap interface and mechanical properties of joints were analyzed. The results showed that under the addition  of Zn interlayer, the  interface structure changed from continuous diffusion layer composed of Fe- Al intermetallic compounds to the uneven and segmented diffusion layer which was composed of  the different combination of Fe-Al, Fe-Zn or Al-Zn intermetallic compounds. The largely-changed  composition and thickness of uniform and segmented diffusion layer at the lap interface played a  significant role in the joint strength. Compared with the joint without Zn, the highest tensile shear  load of the joint with 0.1mm Zn interlayer was 6.26kN increasing by 41.3%. All the joints exhibited  interfacial shear fracture, and the  effective diffusion bonding outside the stir zone  at the  lap  interface  extended the propagation path of crack.

This  paper presents considerable scientific value. It should be revised before publication.

The last paragraph in the introduction section should be carefully describe the main novelty of the paper.

What is the source of Table 1.

 The properties of 301L steel and aluminum alloy should tabulated.

The novelty of the paper should be discussed in the introduction.

The introduction should be supported by discussing the use  artificial intelligence tools, such as adaptive neuro-fuzzy inference system integrated, random vector functional link, marine predators algorithm and  harris hawks optimizer, inn modeling friction stir welding .

The DOE technoque should be described.

The conclusion should be carefully rewritten.

How did the authors select the welding variables?

An experimental set up photo should be added.

The specimen no. should be included in Figure 7.

What is the role of Zn in enhancing the mechanical properties of the joint?

Author Response

Point 1: The last paragraph in the introduction section should be carefully describe the main novelty of the paper. 

Response 1: According to your suggestion, the last paragraph in Section 1 has been changed to carefully describe the research contents of this study. Please check!

Point 2: What is the source of Table 1.

Response 2: By X-ray fluorescence, the chemical compositions of Al-Mg-Si aluminum alloy were obtained, which were shown in Table 1. Thanks!

Point 3: The properties of 301L steel and aluminum alloy should tabulated.

Response 3: The mechanical properties of these two metals have been listed in Table 1 and Table 2 in Section 2. Please check!

Point 4: The novelty of the paper should be discussed in the introduction.

Response 4: The last paragraph of Section 1 has been redescribed carefully to discuss the novelty of this study. Please check!

Point 5: The introduction should be supported by discussing the use artificial intelligence tools, such as adaptive neuro-fuzzy inference system integrated, random vector functional link, marine predators algorithm and harris hawks optimizer, inn modeling friction stir welding. 

Response 5: Paragraph 3 of Introduction has been revised to support this study by discussing the use artificial intelligence tools. Please check!

The added references are as follow:

  • Sr, A.; Rkrs, B.; Klak, A. Artificial intelligent approach for process parameters modeling of friction stir processing. Mater Today Proc, 2021, 43, 326-334.
  • Taher, A.S.; Mohamed, A.; Ammar, H.E.; Zhou, X. Modeling of friction stir welding process using adaptive neuro-fuzzy inference system integrated with harris hawks optimizer. J. Mater Res Technol, 2019, 8(6), 5882-5892.
  • Waheed, S.; E.; Emad I.G.;Essam B. M.; Ammar H.E. A new fine-tuned random vector functional link model using hunger games search optimizer for modeling friction stir welding process of polymeric materials. J. Mater Res Technol, 2021, 14, 1482-1493.

Point 6: The DOE technoque should be described.

Response 6: Thank for your comment. In fact, this study aims at the influence mechanism of the Zn interlayer on the microstructure and mechanical properties of Al/steel joint. The DOE is to further optimize the process parameters to obtain the higher performance joint. The DOE will be an experimental design method to improve the mechanical properties of Al/steel joints under different welding parameter combinations, but it is not suitable in this study. Certainly, how the DOE technique is important has been added in Paragraph 4 of Section 3.2 in the revision. Please check.

The added references are as follow:

  • Zuo, Y. Y.; Kong, L. P.; Liu, Z. L.; Lv, Z.; Wen, H. J. Process parameters optimization of refill friction stir spot welded Al/Cu joint by response surface method. Trans Indian Inst Met, 2020, 73, 2975-2984.
  • Chitturi, V.; Pedapati, S. R.; Awang, M. Effect of tilt angle and pin depth on dissimilar friction stir lap welded joints of aluminum and steel alloys. Mater,2019, 12(23), 3901.

Point 7: The conclusion should be carefully rewritten.

Response 7: The conclusion has been carefully rewritten in the revision. Please check!

Point 8: How did the authors select the welding variables?

Response 8: According to references listed in the following, the welding parameters were selected. The corresponding analysis has been added in Section 2. Please check!

  • Mahto, R. P.; Kumar, R.; Pal, S. K.; Panda, S. K. A comprehensive study on force, temperature, mechanical properties, and micro-structural characterizations in friction stir lap welding of dissimilar materials (AA6061-T6 & AISI304). J Manuf Process, 2018, 31(JAN.), 624-639.
  • Mahto, R. P.; Bhoje, R.; Pal, S. K.; Joshi, H. S.; Das, S. A study on mechanical properties in friction stir lap welding of AA 6061-T6 and AISI 304. Mater Sci. Eng A, 2016, 652(JAN.15), 136-144.
  • Ratanathavorn, W.; Melander, A. Influence of zinc on intermetallic compounds formed in friction stir welding of AA5754 aluminum alloy to galvanised ultra-high strength steel. Sci. Technol. Weld. Joining, 2017, 22(8), 1-8.

Point 9: An experimental set up photo should be added.

Response 9: The photographic view of the weld process and the welded joint have been added in Figure 1. Please check!

Point 10: The specimen no. should be included in Figure 7.

Response 10: The Figure 7 has been modified, and the specimen number has been added in Figure 7. Please check!

Point 11: What is the role of Zn in enhancing the mechanical properties of the joint?

Response 11: The addition of Zn interlayer changed the structure and composition of the diffusion layer at lap interface, and improved the atom diffusion. After adding the Zn interlayer, the diffusion layer between the aluminum and steel sheets was changed from the continuous distribution to the uneven and segmented distribution. In addition, compared to the single Fe-Al IMCs in the diffusion layer of Zn-not-added joint, the Fe-Zn IMCs and Al-Zn eutectic were formed in the Zn-added joint. These results were beneficial to enhancing the mechanical properties of the Al/steel joint.

Reviewer 4 Report

The abstract is good

Why was selected Al-Mg-Si alloy and 301 L and not other materials ? I mean other grades ?

The chemical composition from Table 1 was measured by authors or the data was taken from supplier or literature ?

“with the sandpaper” which grade ?

What you suggest is not actually a diffusion bonding is rather a pressure bonding !

“metallographic sandpapers” is presented in an incomplete manner ! please provide all the steps !

Three tensile shear specimen is very unlikely to provide statistical meaning !

Figure 5 is not clear what the authors want to shows as normally I expected to see how the intermetallic region is formed, now is poor presented

Standard deviation for Figure 7 is required

Figure 8 c and d are poor discussed or even not presented at all

Author Response

Point 1: The abstract is good. 

Response 1: Thank you very much for your comment on the Abstract. At the same time, we also made some amendments to the Abstract.

Point 2: Why was selected Al-Mg-Si alloy and 301 L and not other materials? I mean other grades?

Response 2: The steel and Al-Mg-Si alloy play a vital role in the modern transportation manufacturing field. Therefore, the application of Al-Mg-Si alloy and stainless steel were selected and some further discussions have been added in Paragraph 1 of Introduction. Please check!

Point 3: The chemical composition from Table 1 was measured by authors or the data was taken from supplier or literature? 

Response 3: By X-ray fluorescence, the chemical compositions of these two base materials were obtained. Thanks.

Point 4: “with the sandpaper” which grade ?

Response 4: The sizes of metallographic sandpapers used in this study were 180#, 400#, 800#, 1200# and 2000#, respectively. The corresponding content has been added in Paragraph 3 of Section 2. Please check!

Point 5: What you suggest is not actually a diffusion bonding is rather a pressure bonding! 

Response 5: Thank for your comment. In this study, because the pin does not plunge into the steel sheet, the Al and steel is bonded by the atom diffusion of Al, Fe or Zn elements, and the bonding strength of joint is determined by the degree of atomic diffusion. Therefore, the diffusion bonding is used to define the bonding mechanism of the Al/steel lap interface.

Point 6: “metallographic sandpapers” is presented in an incomplete manner! please provide all the steps!

Response 6: The treatment of metallographic specimens has been rewritten in Paragraph 3 of Section 2. Please check!

Point 7: Three tensile shear specimen is very unlikely to provide statistical meaning!

Response 7: For many reported references [1, 2] listed in the following, the average value of three shear tensile specimens for each parameter combination was used as the final tensile shear load. In the revision, the corresponding analysis has been added in Paragraph 3 of Section 2 according to your comments. Thanks.

  • Mahto, R. P.; Kumar, R.; Pal, S. K.; Panda, S. K. A comprehensive study on force, temperature, mechanical properties, and micro-structural characterizations in friction stir lap welding of dissimilar materials (AA6061-T6 & AISI304). J Manuf Process, 2018, 31(JAN.), 624-639.
  • Movahedi, M.; Kokabi, A. H.; Reihani, S. M. S.; Cheng, W.J.; Wang, C.J. Effect of annealing treatment on joint strength of aluminum/steel friction stir lap weld. Mater Des, 2013, 44(FEB.), 487-492.

Point 8: Figure 5 is not clear what the authors want to shows as normally I expected to see how the intermetallic region is formed, now is poor presented.

Response 8: For the Figure 5, we intend to discuss the structure and compositions of diffusion layer in region C, and analyse the formation process of this diffusion layer. In the revision, the structure and compositions of diffusion layer in region C have been discussed in detail, and the formation process of this diffusion layer has been modified carefully in Paragraph 5 of Section 3.1. Please check!

Point 9: Standard deviation for Figure 7 is required.

Response 9: Figure 7 has been modified, which presents the tensile shear load-displacement of every specimen. Please check!

Point 10: Figure 8 c and d are poor discussed or even not presented at all.

Response 10: According to your suggestion, we have re-analysed and re-discussed the Figures 8c and d. Please check!

Round 2

Reviewer 1 Report

All comments are addressed. 

Reviewer 2 Report

Excellent work - all issues raised have been properly addressed. I did not know that there was a word limit on the length of the abstract. This makes sense then. All the best for this publication.

Reviewer 4 Report

-